# Radiographic Analysis of Grammont-Style and Lateralized Reverse Shoulder Arthroplasty in Gleno-Humeral Osteoarthritis

**Giovanni Merolla [1,2,\*], Giuseppe Sircana [1], Antonio Padolino [1], Francesco Fauci [1], Carlo Alberto Augusti [1], Marco Saporito [1] and Paolo Paladini [1]**

[1]    Shoulder and Elbow Unit, Cervesi Hospital, AUSL Romagna, 47841 Cattolica, Italy
[2]    Biomechanics Laboratory, Cervesi Hospital, AUSL Romagna, 47841 Cattolica, Italy
[\*]   Correspondence: giovannimerolla2010@gmail.com; Tel.: +39-0541-966382

**Abstract:** Reverse shoulder arthroplasty (RSA) has transformed the management of shoulder pathologies, including cuff tear arthropathy and osteoarthritis. The innovative design principles of RSA, such as the medialization and inferiorization of the joint center of rotation, distalization of the humerus, and a semi-constrained construct, enable effective deltoid compensation for rotator cuff deficiency. The Grammont-style RSA demonstrated excellent clinical outcomes. However, complications like instability and scapular notching prompted the exploration of lateralized designs. The radiographic evaluation of RSA is paramount for understanding the biomechanics of the implant and to foresee possible complications. Radiographic assessments encompass glenoid and humeral component positions, identifying features like scapular notching, radiolucent lines, heterotopic ossifications, bone adaptations, and humeral lengthening. Lateralized designs alter muscle moment arms and improve deltoid efficiency, influencing abduction and adduction mechanics. Despite the reduction in scapular notching, lateralized RSA introduces new challenges, such as increased risk of scapular spine and acromial fractures. Understanding the radiographic features and biomechanics of lateralized RSA is crucial for optimizing patient outcomes and mitigating potential complications.

**Keywords:** reverse shoulder arthroplasty; RSA lateralization; postoperative radiology





## 1. Background

Originally indicated for proximal humerus fractures, shoulder arthroplasty applications have been expanded to encompass osteoarthritis, eliciting high patient satisfaction. However, poor results were obtained in the treatment of cuff tear arthropathy due to the inherent limitations of this prosthesis in restoring joint biomechanics.

Reverse prosthesis was proposed as an innovative device to address the biomechanical challenges risen by rotator cuff deficiency. After the failures of pioneering constrained design proposed by Neer, the real game-changer was reverse prosthesis, conceived by Paul Grammont in 1985 [1]. This design featured a marked medialization of the joint center of rotation (CoR), thereby mitigating shear stress on the glenoid in comparison to preceding models. Moreover, it increased the lever arm of the deltoid muscle, allowing for elevation strength recovery [1–3]. The Grammont prosthesis warranted excellent clinical outcomes and a low revision rate [4,5]. These favorable results led surgeons to expand its indications from exclusively addressing rotator cuff tear arthropathy to encompassing fractures and their sequelae, primary osteoarthritis, avascular necrosis, post-traumatic osteoarthritis, and even cases of irreparable rotator cuff tears [6–8]. Over the past decade, the use of reverse shoulder arthroplasty (RSA) has experienced a remarkable increase, nearly doubling the number of procedures performed [9]. Presently, RSA constitutes 57.43% of all shoulder arthroplasties performed in the United States [10], and this number is bound to increase.

## 2. Gleno-Humeral Deformity in Primary Osteoarthritis and Cuff Tear Arthropathy

Degenerative changes in the glenohumeral joint are found in up to 17% of patients with shoulder pain, a patient group that has tripled in the last 40 years [11,12]. Two imaging studies using the Samilson–Prieto classification showed glenohumeral osteoarthritis (GHOA) prevalence rates in the middle-aged and elderly to be as high as 17–19%, while bilateral disease was identified in 3.1–7.7% of the population [13,14]. Prevalence rates of secondary GHOA were reported at only 1.3–1.7%, making age-related primary GHOA 10 times more common than secondary GHOA [13,14]. However, GHOA due to a specific identifiable cause has been shown to be significantly more common in patients aged less than 50 [15]. Although symptomatic GHOA is not as common as osteoarthritis of the hip and knee joints, it can be just as debilitating due to the functional importance of the upper limbs [16,17].

Etiology of GHOA is poorly understood and most of our knowledge derives from advancements made in understanding OA in general. Many theories regarding the development of GHOA were proposed, but ultimately we can consider OA a common endpoint of a heterogeneous group of disorders that lead to degenerative joint damage [18]. Academic classification of GHOA divides it into primary or idiopathic, and secondary, when a specific cause can be identified. The pathogenesis of joint damage, seen as part of a common pathological process, is influenced by multiple factors [19]. These can be divided into non-specific and specific factors as well as into systemic and local factors. Joint damage develops from the interplay between these factors, where local or systemic factors, or non-specific or specific factors, may dominate [18,20]. Disease progression is, however, typically affected by a combination of genetic, behavioral, and environmental factors [14,21–25].

GHOA caused by traumatic or degenerative rotator cuff tears displays three characteristic changes. These are rotator cuff insufficiency, cranial migration of the humeral head, and subsequent radiographic degenerative changes. Radiographs typically show bony erosion of the superior glenoid, resulting in acetabularization of the coracoacromial arch and rounding off of the humeral greater tuberosity [26]. Cuff tear arthropathy (CTA) is seen more commonly in women and mostly affects the shoulder of the dominant arm [17].

Primary GHOA can be classified according to Samilson and Prieto [27]. It classifies GHOA in three grades: grade 1, inferior humeral or glenoid exostosis, or both, measuring less than 3 mm in height; grade 2, inferior humeral or glenoid exostosis, or both, between 3 and 7 mm in height, with slight glenohumeral joint irregularity; and grade 3, inferior humeral or glenoid exostosis, or both, more than 7 mm in height, with narrowing of the glenohumeral joint and sclerosis. Minor modifications to this classification were proposed by Allain [28] and Gerber [29]. Another radiographic classification was developed by Guyette [30], dividing GHOA into four grades: grade 0, no appreciable signs of arthritis; grade 1, mild sclerosis and/or a small osteophyte less than 2 mm on only one side of the joint; grade 2, large marginal osteophytes or osteophytes on more than one side or surface of the joint and joint space narrowing and/or the presence of cysts; and grade 3, joint surface destruction, bone on bone, joint space narrowing, and/or loose bodies.

Classification of arthritic changes due to rotator cuff deficiency is typically performed according to Hamada et al. [31]. Their classification divides arthropathy into five grades: grade 1, AHI greater than 6 mm; grade 2, AHI 5 mm or less; grade 3, presence of acromial acetabulization; grade 4, narrowing of the glenohumeral joint along with grade 3 features; and grade 5, humeral head collapse. Afterwards, Walch et al. [32] divided the Hamada grade 4 into two subtypes: grade 4A, with narrowing of the glenohumeral joint without subacromial acetabulization, and grade 4B, with narrowing of the glenohumeral joint in the setting of subacromial acetabulization.

A more precise evaluation of glenohumeral changes, paramount in preoperative planning, requires the use of a CT scan. The basic measurement of glenoid morphology includes the inclination and the version in coronal and axial planes, respectively [33,34]. Unfortunately, these values are greatly dependent on scapular positioning and CT plain definition, resulting in measurement error [35,36]. Using a standardized position of the

patient in the CT scan gantry and applying 3D-based reconstruction and planes reformatting significantly reduces measurement errors [35,37]. Walch et al. [38] developed a classification system to describe glenoid morphology in cases of primary glenohumeral osteoarthritis. Since that classification system was first presented, several authors have commented on the interobserver and intraobserver reliability of the classification, with varying results [39–41]. The main limitation of the original Walch classification was the use of traditional 2D CT scans, which have since been found to portray glenoid version less reliably than 3D reconstructions that analyze the scapula as a free body, as reported above [42–45]. The original classification includes five categories of glenoid patterns: A1—centered humeral head, minor erosion; A2—centered humeral head, major central glenoid erosion; B1—posterior subluxated head, no bony erosion; B2—posterior subluxated head, posterior erosion with biconcavity of the glenoid; and C—dysplastic glenoid with at least 25° of retroversion regardless of erosion [38]. Bercik et al. [46] proposed several modifications to this original classification system, suggesting the addition of the "B3" and "D" glenoids and a more precise definition of the A2 glenoid (Figure 1).

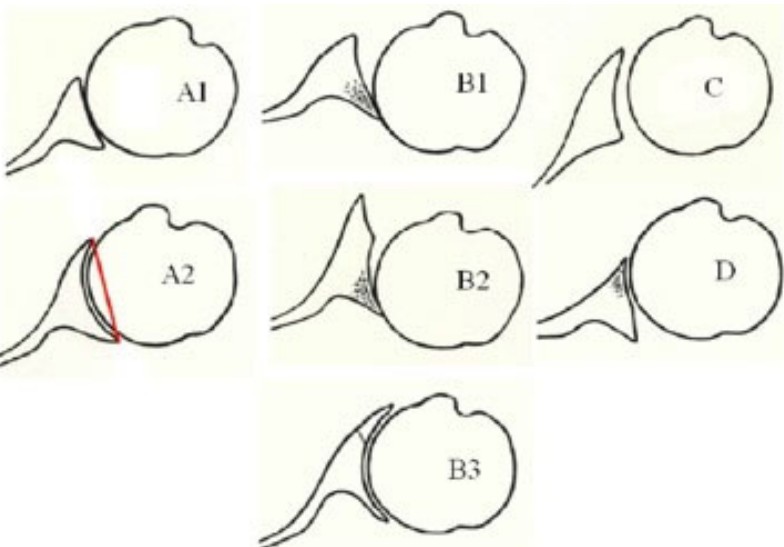

**Figure 1.** Representation of glenoid morphology in osteoarthritis according to Walch et al. as modified by Bercik et al. [46].

They defined the B3 glenoid as monoconcave and posteriorly worn, with at least 15° of retroversion or at least 70% posterior humeral head subluxation, or both. The B3 glenoid with posterior subluxation without significant retroversion differs from the B1 due to the presence of posterior wear. They defined the D glenoid as one with any level of glenoid anteversion or with humeral head subluxation of less than 40% (i.e., anterior subluxation). The definition of the A2 glenoid "cupula" was also updated to describe glenoids in which a line drawn from the anterior to posterior rims of the native glenoid transects the humeral head. This contrasted with the A1 glenoid, in which a line drawn from the anterior to the posterior rim of the native glenoid does not transect the humeral head. Lastly, they clarified the C glenoid to be a dysplastic glenoid with at least 25° of retroversion "not caused by erosion". Walch classification evaluates glenoid morphology on the transverse plane, but coronal plane deformity requires assessment as well. Superior glenoid wear is common in patients with CTA and results from the progressive erosion of the glenoid by the superiorly migrated humeral head [47,48]. About 37.5% of patients with rotator-cuff-deficient shoulders had some degree of glenoid wear, and more advanced CTA has been associated with superior glenoid wear [47,49] (Figure 2).

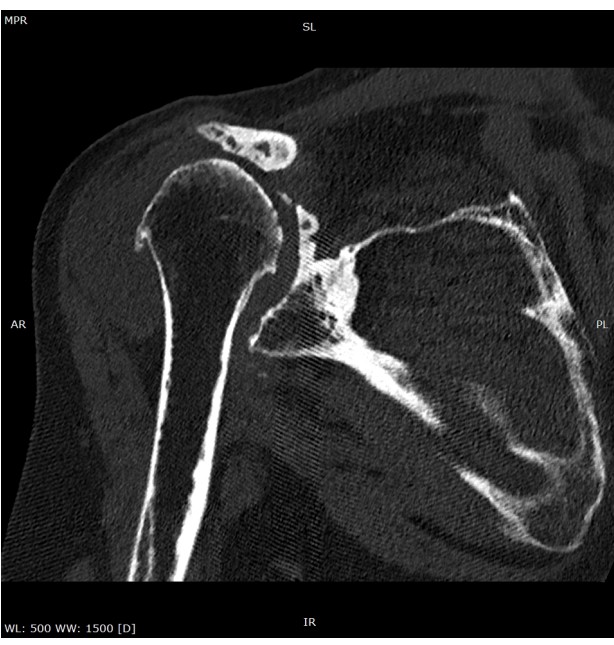

**Figure 2.** Superior migration of the humeral head in CTA.

Favard classified glenoid wear with rotator cuff tear arthritis based on the location and extent of erosion at the superior and inferior aspects of the glenoid [50]. The Favard superior glenoid wear classification system includes types E0, no glenoid wear; E1, concentric erosion of the glenoid; E2, erosion limited to the upper part of the glenoid; E3, erosion extending to the inferior part of the glenoid; and E4, erosion predominantly located at the inferior part of the glenoid (Figure 3).

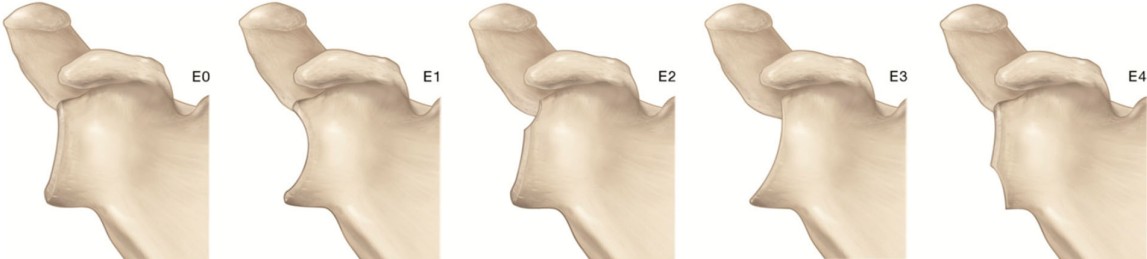

**Figure 3.** Representation of glenoid wear in cuff tear arthritis according to Favard et al. [50].

Recent studies have aimed to quantify the direction and amount of glenoid erosion using 3D reconstructions of CT scans. A 3D reconstruction is crucial for planning the correction of posterior glenoid bone loss and pathologic retroversion in order to restore the native joint line [51,52]. The use of 3D CT reconstruction allowed for the identification of multiplanar glenoid deformity, and consequently the position of the implant components that prevent overmedialization of the joint line. Three-dimensional CT imaging can assist shoulder surgeons to choose between asymmetric reaming, glenoid bone grafting, and augmented glenoid components to address excessive retroversion and glenoid bone loss [53–55]. Otto et al. described three different patterns of multiplanar glenoid wear in type B2 biconcave glenoids [56]. They found that the most common pattern of erosion was in the posterior–central direction, with the remaining cohort falling into either the posterior–inferior or posterior–superior direction. Across all deformities, Knowles et al. reported that the line of glenoid erosion in B2 glenoids was directed toward the posterior–inferior quadrant, with the orientation of bone loss directed toward the 8 o'clock position; in addition, the radius of the curvature for the neoglenoid was flatter than the paleoglenoid [52]. Otto et al. found that the direction deviated in many cases from the 8 o'clock position [56].

Overall, this information may assist surgeons in addressing the technical factors associated with glenoid resurfacing of the B2 erosion pattern and manufacturers in the fabrication of implants that can better address commonly seen glenoid deformities.

### 3. Biomechanics of Reverse Shoulder Arthroplasty

Grammont based its innovative reverse shoulder arthroplasty on four fundamental biomechanical principles [1,57,58]:

- Medialization and inferiorization of the joint CoR, achieved through medialization of the glenoid and the humerus with a 145° neck–shaft angle to increase the deltoid lever arm.
- Setting the CoR at the bone–prosthetic glenoid interface, thus reducing shear forces on the metal back.
- Distalization of the humerus, thus tensioning the deltoid to recover strength even from the onset of the motion.
- Semi-constrained design, obtained with a convex glenosphere and a concave humeral cup with the same curvature radius, creating a ball and socket joint and providing a stable fulcrum and ensuring static stability.

With these biomechanical adaptations, deltoid acts as a forward elevator and abductor, effectively compensating for the non-functional rotator cuff. The resultant RSA demonstrated remarkable clinical outcomes, encompassing an average active flexion of 134°, abduction of 116°, and external rotation of 36°. This was accompanied by a notable mean enhancement of the Constant score by 37 points [59] (Figure 4).

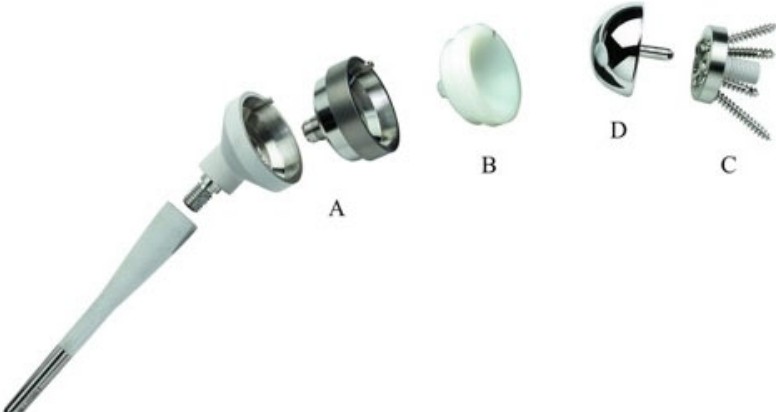

**Figure 4.** Grammont reverse design of shoulder arthroplasty. The humeral component included: (**A**) a distal stem with a metaphysis and an optional humeral spacer; (**B**) a centered (standard) and retentive constrained humeral insert, with variable height (+6 mm, +9 mm, and +12 mm). The glenoid component included: (**C**) a hydroxyapatite-coated baseplate (25 and 29 mm) with a central post and 4 self-tapping screws (2 compression screws and 2 multidirectional locking screws), and (**D**) centered and eccentric (+2 mm and +4 mm) glenospheres (Stryker, Kalamazoo, MI, USA).

Regrettably, an overall complication rate of 9.4% was encountered [59]. Despite the satisfactory outcomes, some flaws became apparent in this design: medialization could lead to detensioning of the remaining rotator cuff [60], resulting in loosing potential strength and ROM and reducing deltoid wrapping [61]. These factors predisposed to instability [62], a primary complication with a reported incidence rate of 4% [63]. Additionally, medialization alters the contour of the shoulder [64,65] due to less offset and increased arm length [64]. Medialization also decreases anterior and posterior deltoid tension, thereby reducing rotation strength [60]. The combined effects of glenoid medialization and an increased neck–shaft angle leads to scapular notching in up to 42.5% of cases, which can ultimately result in polyethylene wear and glenoid loosening [66]. In efforts to address or mitigate

these limitations, a lateralized or, to be biomechanically more accurate, less medialized design was proposed (Figure 5).

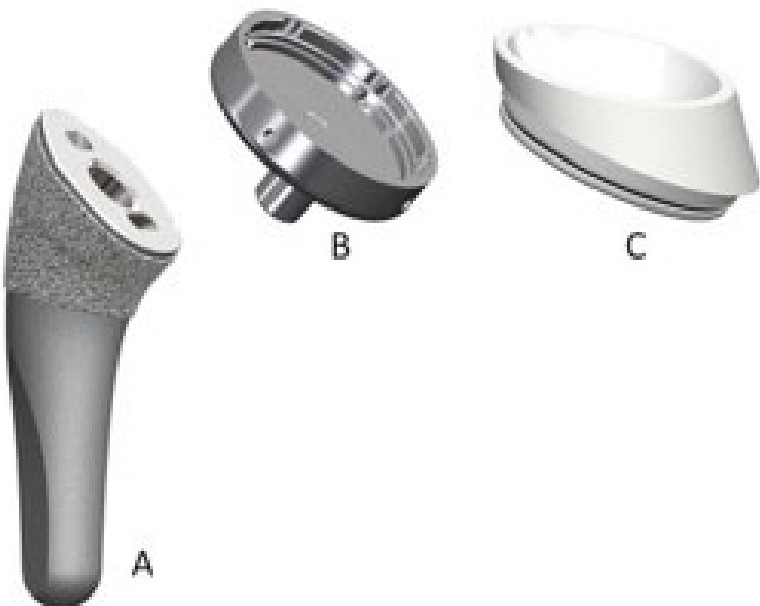

**Figure 5.** Short-stem lateralized humeral component of reverse shoulder arthroplasty. (**A**) A hydroxyapatite-coated curved short-stem 132.5° inclination; (**B**) an eccentric (+1.5 mm and +3.5 mm) reverse tray; and (**C**) an asymmetric polyethylene insert (thickness + 6 mm and + 9 mm) with 12.5° inclination (Stryker, Kalamazoo, MI, USA).

Lateralization can be achieved through the modification of the glenoid side, the humeral side, or both. Glenoid lateralization strategies include alterations in the scapular neck [67], in the baseplate [68], or in the glenosphere [69]. Lateralization at the scapular neck is obtained through the insertion of a graft between the glenoid surface and the glenoid component (Bio-RSA). The graft can be cylindrical [67] or asymmetric [53] in order to accommodate glenoid morphology and wear. A recent metanalysis [70] revealed superior results for the Bio-RSA over standard RSA in terms of the Constant score and subjective shoulder value (SSV). While improvements in range of motion (ROM) were noted in forward flexion, external rotation, and internal rotation, statistical significance in superiority was not established. Lateralization can be obtained by increasing the thickness of the glenoid implant, or by the choice of a glenosphere whose thickness is higher than its radius, namely utilizing a more-than-hemispheric design. Irrespective of the approach, the result is the lateralization of joint CoR relative to the glenoid surface. On the humeral side, lateralization can be realized by reducing the neck–shaft angle to either 145° or 135°, or by employing onlay systems or curved stems.

The lateralized RSA design has several biomechanical advantages. First, it increases tension on the remaining posterior rotator cuff tendons, resulting in enhanced external rotation strength [61]. Additionally, the lateralization of the CoR has multiple positive effects. By moving the CoR away from the scapular bony surfaces, lateralization increases the impingement-free range of motion (ROM) and reduces the risk of notching [71–73]. This design alteration also promotes better deltoid wrapping by shifting the deltoid force vector more medially across the humerus. This, in turn, increases joint reaction forces and enhances overall stability [61,74]. In fact, studies have reported a notable decrease in instability rates, dropping from 4.0% to 1.3% with the adoption of a lateralized design [63]. Moreover, lateralization of the CoR reduces the number of deltoid fibers situated laterally to the CoR. This has the beneficial effect of restoring the functions of anterior and posterior deltoid fibers as rotators and elevators [58]. For the same reason, this design modification helps in achieving a near-normal contour of the shoulder [58].

However, the pursuit of lateralization is not devoid of risks, despite its biomechanical advantages. Lateralization of the CoR increases stress at the glenoid bone implant interface, and the increase in stress is proportional to the amount of lateralization [75] (Figure 6).

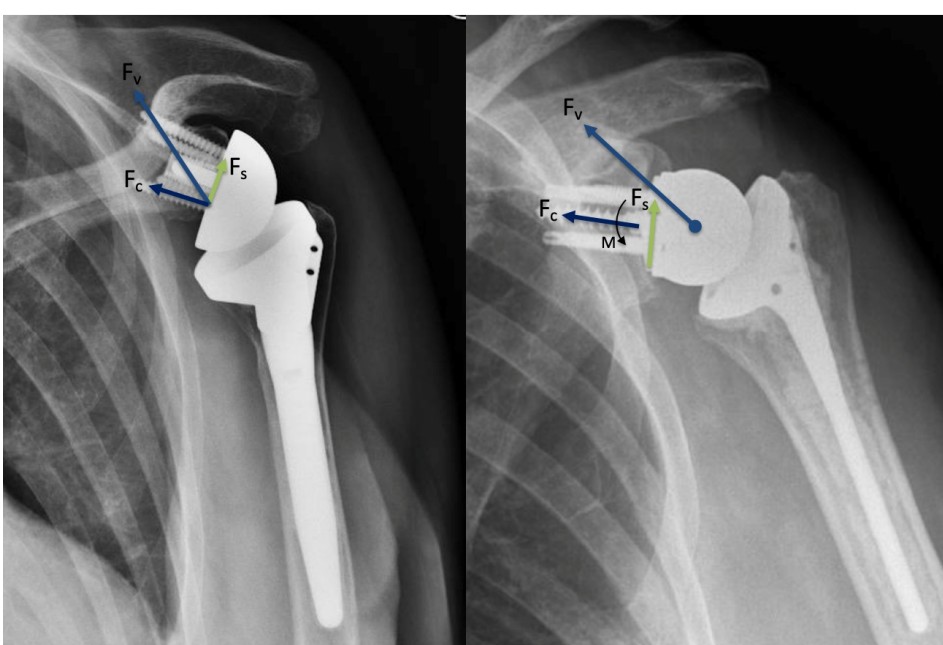

**Figure 6.** RSA acting with a fixed CoR with compressive ($F_c$) and shear ($F_s$) components of the resultant force vector ($F_v$). B. CoR lateralization increases the lever arm length which decreases compressive forces, increases destabilizing shear forces, and creates a new moment (M) at the glenoid–implant interface [3]. RSA: reverse shoulder arthroplasty; CoR: center of rotation.

Notably, prosthetic lateralization induced lower micromotion at the bone–implant interface compared to graft lateralization [75]. This suggests that 10 mm of prosthetic lateralization is tolerable, whereas as little as 5 mm of graft lateralization exhibited significant micromotion [75]. Furthermore, lateralization of the CoR leads to a reduction in the deltoid moment arm, increasing, as a consequence, the deltoid muscle force required for abduction [76].

A comprehensive classification of implant lateralization was devised by Werthel et al. [57]. This classification separately assessed glenoid and humeral offsets across various available implants, employing the DePuy Delta III as a reference (Figure 7).

As per this classification, a glenoid was categorized as medialized if its lateral offset deviated by less than 5 mm from that of the Delta III, and as lateralized if the offset exceeded 5 mm. Similarly, a humerus was classified as medialized if its lateral offset was less than 5 mm higher than that of the Delta III, minimally lateralized if it was 5–9 mm higher, and lateralized if it was 10–14 mm higher. By combining glenoid and humeral lateralization, a global implant lateralization categorization could be derived. Accordingly, implants were classified as medialized (global offset < 5 mm of Delta III), minimally lateralized (offset 5–10 mm of Delta III), lateralized (offset 10–15 mm of Delta III), highly lateralized (offset 15–20 mm of Delta III), and very highly lateralized (offset > 20 mm of Delta III). Commercially available implants encompassed a wide range of lateral offsets, spanning from 13.1 mm of Grammont to 35.8 mm in the most lateralized design.

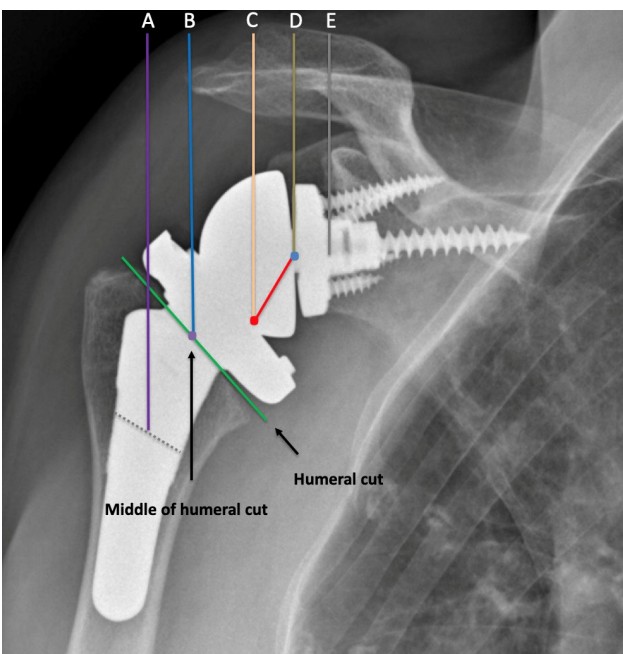

**Figure 7.** Radiographic references to measure humeral and glenoid offset in lateralized RSA (global lateral offset), as described by Werthel et al. [19]. Line A is the vertical line passing through the middle of the diaphysis of the humeral stem. Line B is the horizontal line passing through the middle of the surface of the humeral implant at the level of the humeral cut. Line C is the vertical line passing through the "pivot point" defined as the deepest point of the articular surface of the humeral insert measured perpendicular to the surface of the humeral insert. Line D is the vertical line passing through the center of rotation of the joint. Line E is the vertical line passing through the bone–glenoid baseplate interface. Humeral lateral offset (distance AC) was defined as the sum of the humeral stem offset (distance AB) and of the humeral insert offset (distance BC). Glenoid lateral offset (distance CE) was defined as the sum of the "perceived radius of the glenosphere" (distance CD) and of the center of rotation offset (distance DE).

## 4. Radiographic Evaluation of Grammont-Style Reverse Arthroplasty

The shoulder series to evaluate the components position in RSA are fundamentally composed of two orthogonal views of the glenohumeral joint, including the entire scapula (anterior–posterior [AP] Grashey and outlet views) and axillary view [77–79] (Figure 8).

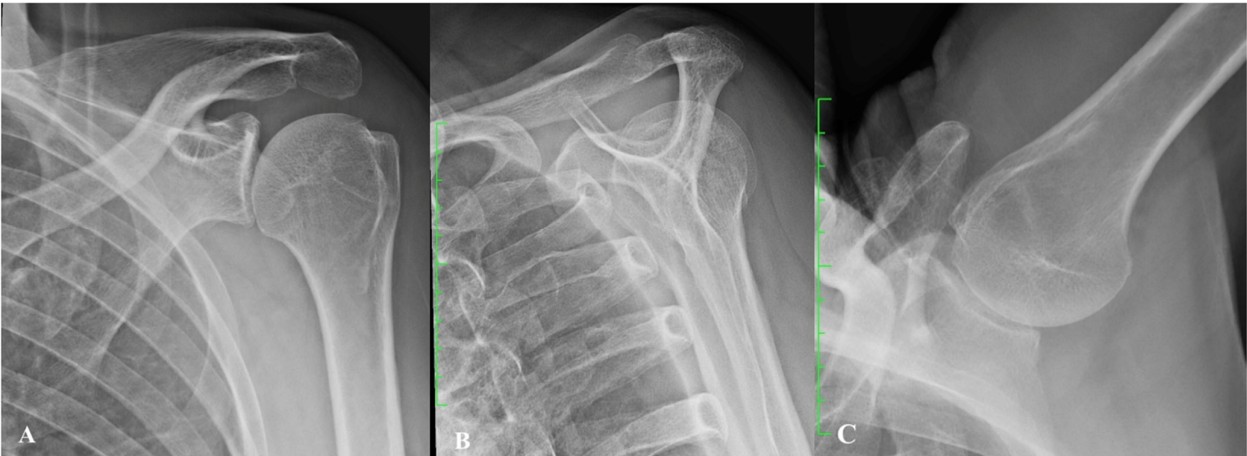

**Figure 8.** Standard radiograph series to assess the shoulder joint. (**A**) True AP view, (**B**) outlet view, (**C**) axillary view. AP: anterior–posterior.

Humeral and glenoid components of RSA should be assessed in the early postoperative phase and at later follow-ups.

Glenoid component radiographic features include glenoid component height and tilt, scapular notching, radiolucent lines and loosening, heterotopic ossifications, and scapular fractures.

Glenoid component height is defined as the distance between the inferior edge of the baseplate and the inferior edge of the bony glenoid. It is categorized as adequate if there is an inferior overhang, high if the glenosphere grazes the inferior edge of the glenoid, and excessively high if it extends beyond [80]. Inferior offset of the glenosphere has been linked to improved outcomes [81]: inferior glenoid positioning decreases notching and increases ROM both in elevation and in rotation [71,82,83]. Glenoid inclination can be easily measured as the angle between the floor of the supraspinatus fossa and glenoid component [79]. This method, however, does not allow for an accurate evaluation of the correct implantation of an RSA. To achieve a more reproducible measurement, β-angle was proposed by Maurer et al. [84]. This is defined as the angle comprised between the floor of the supraspinatus fossa and a line connecting the upper and lower pole of the glenoid. It evaluates global glenoid inclination, not taking into account the fact that RSA is usually placed in the lower part of the glenoid, that may show a different inclination caused by wear [85]. To overcome this issue, the RSA angle was introduced (Figure 9): this is the angle defined between the perpendicular to the supraspinatus fossa line passing through the distal pole of the glenoid fossa and the line drawn from the distal pole of the glenoid fossa and the point where the supraspinatus fossa line crosses the glenoid [85]. This angle allows, on preoperative radiographs, an evaluation of the inclination of the inferior glenoid cavity and, on postoperative radiographs, to understand whether the gap has been adequately filled, ensuring correct baseplate inclination [85,86]. Adequate correction of glenoid inclination is paramount in correct RSA positioning: superior inclination is a risk factor for reduced ROM, loosening, and instability [47,87–89].

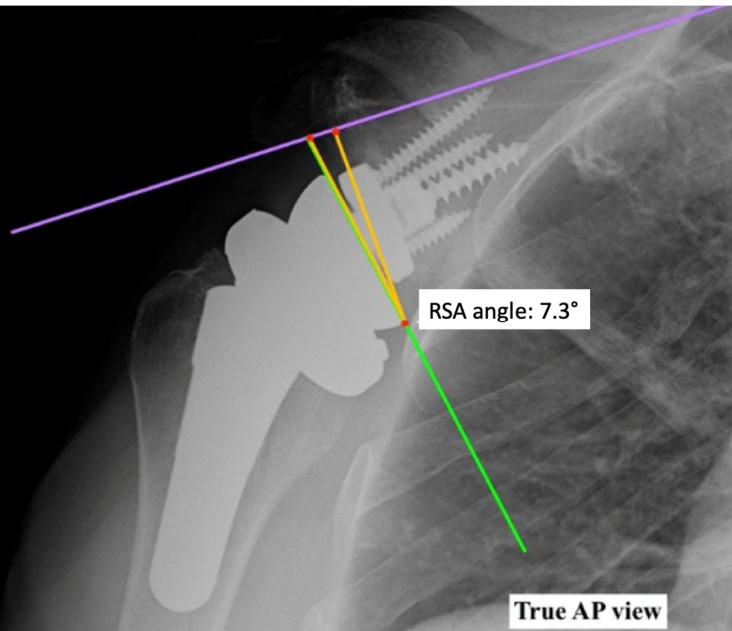

**Figure 9.** The RSA angle is the angle between the inferior part of the glenoid fossa (where the baseplate is implanted) and the perpendicular to the floor of the supraspinatus fossa.

Scapular notching is the paramount complication of Grammont design, with an incidence of 42%. It is classified according to the Nerot–Sirveaux grading system [50] based on the size of the defect evaluated on AP X-Rays. A defect which is confined to the pillar corresponds to grade 1. In grade 2, the defect is in contact with the lower screw; in grade 3,

the defect is over the lower screw; and in grade 4, it extends under the baseplate. The clinical effect of notching is still debated; while severe notching predisposes to glenoid loosening, leading to RSA failure [90], a relationship between notching and clinical outcomes was inconsistently found among different patient cohorts [91,92].

Radiolucent lines are considered as significant when >2 mm. They can be located around the screws, central peg, or baseplate [93]. Their interpretation is complicated, as their presence is not directly linked to the clinical outcome, yet they may be a sing of incipient loosening [93,94]. Loosening may be declared if the glenoid component migrated, as demonstrated by shift, tilt, or subsidence, or if complete radiolucency ≥2 mm was present in every part of the glenoid [93].

Scapular spine or acromial fracture are a relatively common complication, occurring in 2.5% of cases, and are associated with inflammatory arthritis [63]. Acromial fractures are more common than scapular spine fractures. Levy et al. [95] categorized them into three types: type I includes fractures of the midpart of the acromion, involving anterior or middle deltoid origin; type II involves the middle and a portion of the posterior deltoid origin; and type III fractures involve the entire middle and posterior glenoid origin.

Information regarding the humeral component can also be extracted. Evaluation includes stem alignment, radiographic changes around the stem, and subsidence [77].

The alignment of the prosthesis is assessed by measuring the angle between the humeral shaft axis and a line connecting the geometric center of the humeral head to the prosthesis tip. Varus or valgus alignment of the stem changes the resulting prosthetic neck–shaft angle, thus altering lateralization [96]. Stem malalignment is a common postoperative finding with short stems, with a reported prevalence of 19% [97].

Bone remodeling around the stem can be revealed by condensation lines around the tip of the stem [93], cortical thinning and osteopenia, and spot welds around the humeral component [78,93]. An assessment of the changes should be conducted both medially and laterally at one third and two thirds of the stem's length [78] and underneath the stem [77], resulting in five measurement spots (Figure 10).

Other authors [98–100] divide the stem into seven zones, with two comprising the metaphyseal part, four comprising the diaphyseal part of the stem, and one surrounding the stem tip (Figure 10). Changes in these zones are summed up, and adaptations can be rated as none (0–1 changes), mild (2–3 changes), moderate (4–6 changes), or advanced (changes in every one of the five zones or changes with aggressive behavior) [77].

For short stems, the metaphyseal filling ratio can also be evaluated as the ratio between the metaphyseal stem width and metaphyseal humeral width [77].

Postoperative imaging can also provide useful information about implant stability. Many intraoperative methods are counselled for the assessment of implant stability [1,60], but the only quantitative available method is the postoperative evaluation of arm lengthening.

Pre- and postoperative assessment of the AHD allows for obtaining indirect insight into arm lengthening. This distance is measured between the most lateral part of the acromion's undersurface and a line parallel to the top of the greater tuberosity [79,101]. Distance from the AHD line perpendicular to the most lateral portion of the greater tuberosity measures the lateral humeral offset [79,101]. An increased acromion–humeral distance correlates with improved ROM [102]. Nevertheless, arm lengthening can be evaluated using direct methods. Bilateral preoperative and postoperative true anteroposterior scaled radiographs of the humerus, taken in neutral rotation and with the patient standing, are needed. The humeral length, measured along the diaphyseal axis from the epicondylar line to the humeral head or prosthesis top, serves as a reference point. By comparing this with a line perpendicular to the diaphyseal axis passing through the most lateral and inferior acromion point, humeral lengthening can be calculated [103]. Mean humeral lengthening after arthroplasty ranges, in the literature, from 15 mm to 27, with shortening associated with increased instability [103,104]. Humeral distalization greater than 28 mm is linked to enhanced elevation and reduced notching [105].

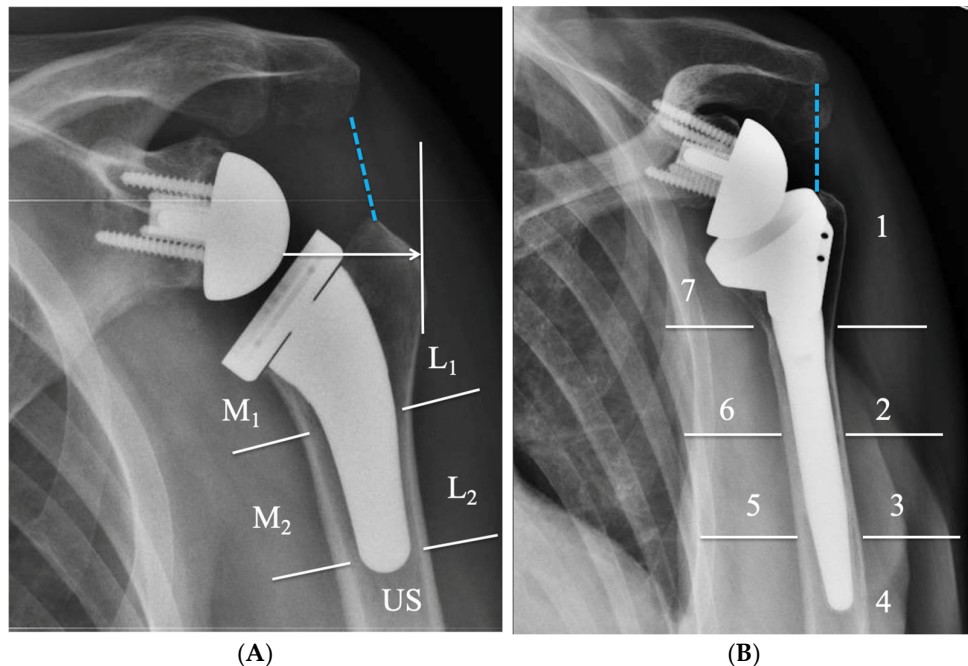

**Figure 10.** Anterior–posterior Grashey view of the 2 reverse prostheses. (**A**) Onlay curved short-stem design. (**B**) Medialized standard-stem design (Grammont design). The curved-stem design increases the humeral offset (white arrow), whereas the acromiohumeral interval (dotted lines) remains unchanged if the humeral tray is placed below the greater tuberosity. Humeral bone remodeling and radiolucency can be assessed in 5 zones of the short stem and in 7 zones of the standard straight stem. M1, medial 1; M2, medial 2; L1, lateral 1; L2, lateral 2; US, under stem.

## 5. Radiographic Features of Lateralized Reverse Arthroplasty

Quantitative radiographic analysis of lateralized RSA is performed using standard views previously described. Glenoid lateral offset (LO) is calculated as the sum of the radius of the glenosphere and of the CoR offset. By increasing the diameter of the glenosphere to the largest available size, glenoid lateralization increases by a mean of only 1.14 mm. This is a limited increase in lateral offset, compared with CoR lateralization or humeral lateralization [57].

LO in RSA changed significantly, passing from a mean of 13.1 mm with a medialized Grammont prosthesis up to 35.8 mm with the current design of lateralized RSA.

Changes in RSA design to achieve lateralization of the CoR have been performed through glenosphere shape modification or baseplate lateralization.

The medialized design of RSA has a resultant force vector that acts through a fixed CoR [1]. When a specific glenosphere design is created to obtain a CoR lateralization, this decreases compressive forces and increases destabilizing shear forces, creating a new moment at the glenoid–implant interface [106] (Figure 6).

When glenoid lateralization is achieved through the baseplate, the CoR falls at the baseplate–glenosphere interface, and this type of design should reduce the risk of micro-motion at the glenoid bone–implant interface. Bone-increased offset RSA (BIO-RSA) was introduced as a new concept of "biologic lateralization". In this procedure, a bone graft, obtained from the removed humeral head or from an allograft, is placed behind the glenoid component, thus thickening the scapular neck. Increasing the length of the scapular neck would lateralize both the CoR and glenoid–implant interface [67].

The CoR is maintained at the glenoid–implant interface, thus minimizing the torque on the glenoid component. Radiographic assessment of bone graft radiolucency and thickness of BIO-RSA allows us to identify bone graft healing and cases of baseplate "at risk" of loosening (radiolucent lines >2 mm).

Radiolucent lines at the interface "glenoid bone-metal" of metallic increased RSA (MIO-RSA) explain the baseplate seating (no radiolucent lines: perfect seating; radiolucent lines <2 mm: incomplete seating; and radiolucent lines >2 mm: loosening).

Radiolucent lines around the glenoid component are assessed in five zones. Additional glenoid radiographic features include scapular notching, bone scapular spurs, and ossifications [107]. Multiple radiolucent lines <2 mm or one RL $\geq$ 2 mm represent risk factors of glenoid component failure.

AP view is used to assess the vertical position of the glenosphere (i.e, glenosphere inferior overhang [GIO]) [108] and glenosphere inclination [84], as described in the previous sections. Insufficient GIO, following a high or flush placement of the glenosphere and a superior inclination of the baseplate (RSA > 5°), is associated with a higher risk of scapular notching. Acromio-humeral interval and lateral humeral offset provide details about the amount of humeral distalization and lateralization; furthermore, these parameters can affect deltoid tension.

Recent research findings introduced the distalization shoulder angle (DSA) and lateralization shoulder angle (LSA) as new radiographic methods to measure the inferior and lateral position of the humerus [109] (Figure 11).

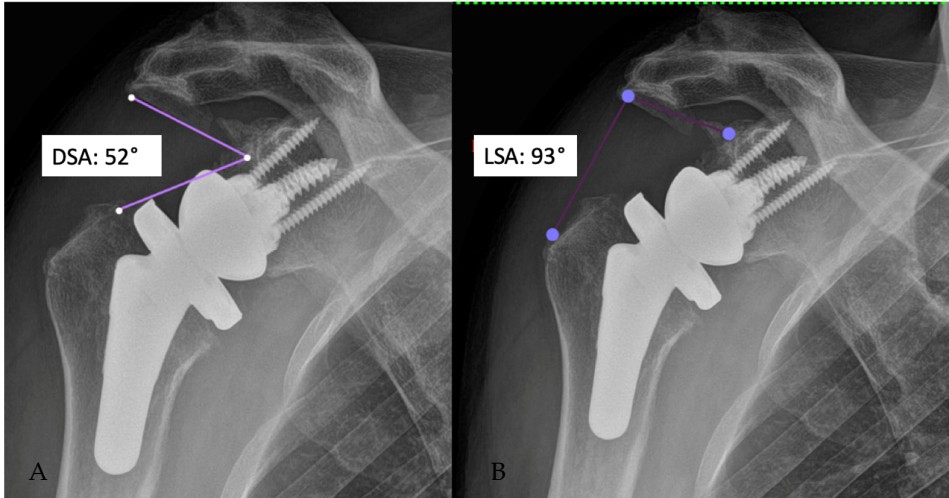

**Figure 11.** Distalization shoulder angle (DSA) (**A**) and lateralization shoulder angle (LSA) (**B**).

The active range of motion after RSA is correlated with specific ranges of LSA and DSA. LSA between 75° and 95° is correlated with an increased active external rotation, and a DSA between 40° and 65° is correlated with increased active anterior elevation [109]. The current design of RSA allows for the performance of a lateralization on the glenoid side, humeral side, or both (global lateralization). The sum of humeral LO and glenoid LO gives global lateralization. Humeral LO is measured as the distance between the vertical line passing through the middle of the humeral stem diaphysis and the vertical line passing through the deepest point of the humeral insert surface. Glenoid LO is computed as the distance between the vertical line passing through the deepest point of the humeral insert surface and the vertical line passing through the glenoid bone–prosthesis interface [57].

The range of global lateralization achieved with an RSA implant varies from 3.3 to 20.9 mm. Current RSA devices allow for glenoid lateralization up to 7.7 with a standard baseplate and up to 8.8 mm with bone or metallic augments. The amount of glenoid lateralization required in RSA is still debated. However, surgeons are aware that they should be cautious with glenoid lateralization beyond +5–10 mm, depending on the amount of loss of medial bone stock, to avoid the risk of scapular facture or brachial plexus nerve palsy [110] (Figure 7).

Radiographic analysis of other established parameters around the humeral component includes radiolucency, condensation lines, cortical thinning, spot weld, subsidence, and

resorption of the tuberosities. These radiographic features assess the stability of the humeral component and the risk of stress shielding on the cortical of the humerus.

## 6. Biomechanical Considerations of Lateralized Reverse Implants

Biomechanical studies of RSA showed that RSA design has effects on muscle moment arms [74].

The improved deltoid efficiency after RSA reduces the force to achieve humeral elevation. Distalization of the CoR has the effect of inducing a much larger adduction through the activity of infraspinatus and subscapularis. Offset of the humerus affects the efficiency of posterior deltoid, while a lateral CoR did not affect the muscle moment arm of external rotators [74]. These biomechanical findings confirm the efficacy of reverse design to improve deltoid lever arm when the CoR is located at the glenoid–bone interface. The use of augmented baseplate preserves glenoid bone stock in case of severe glenoid erosion [111]. Recent studies reported that an augmented baseplate preserves 54% more native bone than a nonaugmented baseplate and results in 4.1 mm of lateralization [110]. Moreover, preserving glenoid bone can prevent the risk of impingement and scapular notching.

Clinical studies demonstrated that RSA with a 135° neck–shaft angle and a lateralized glenosphere enables the better preservation of an external rotation and reduces the rate of scapular notching compared with the classic Grammont design [112]. These clinical outcomes are not in line with other biomechanical findings, demonstrating the poor effects of lateralized design on external rotators muscles [74]. Lateralization in RSA has reduced the risk of notching, almost eliminating it, but it has increased the risk of scapular spine and acromial fractures [113–115]. The effects of lateralized RSA on internal rotation mobility are controversial, and the results of recent clinical studies, assessing different reverse designs, are elusive.

There is substantial agreement, as confirmed by recent clinical studies, that a larger glenosphere size, a posterior offset humeral cup, and an increased inferior glenosphere overhang are associated with excellent outcomes [81].

## 7. Conclusions

In this study, we have explored the radiographic evaluation of RSA with a focus on lateralized designs. Our analysis revealed significant advancements over time in RSA design to enhance the biomechanical efficiency and clinical outcomes. The lateralization of the center of rotation, whether achieved through glenoid or humeral modifications, has shown improvement in deltoid function and overall patient outcomes. Additionally, radiographic parameters such as glenoid offset, humeral offset, and humeral distalization have been identified as critical indicators in assessing the success of RSA procedures. However, it is crucial to acknowledge the limitations in our understanding, including the need for standardized measurement techniques and a more comprehensive understanding of the ideal amount of lateralization. The lack of uniform measurement standards and interobserver variability in assessing radiographic parameters may introduce measurement errors and inconsistencies. Furthermore, the clinical correlation of radiographic findings and patient-reported outcomes requires further investigation to establish stronger associations.

Investigating the long-term clinical implications of lateralized RSA and its impact on patient satisfaction, function, and implant longevity is essential. Additionally, exploring novel designs and materials for RSA implants, along with their radiographic implications, will contribute to the ongoing evolution of shoulder arthroplasty techniques. Ultimately, the integration of advanced imaging modalities and computational modeling could offer a more comprehensive understanding of RSA biomechanics and guide personalized implant selection and surgical planning.

**Author Contributions:** Conceptualization, G.M. and G.S.; methodology, G.M. and G.S.; writing—original draft preparation, G.M., G.S., A.P., F.F., C.A.A. and M.S.; writing—review and editing, G.M. and P.P.; supervision, P.P. All authors have read and agreed to the published version of the manuscript.

**Funding:** This research received no external funding.

**Conflicts of Interest:** The authors declare no conflict of interest.

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
