# Peer review of "Radiographic Analysis of Grammont-Style and Lateralized Reverse Shoulder Arthroplasty in Gleno-Humeral Osteoarthritis"

_prosthesis, doi:10.3390/prosthesis5040075_

Round 1

Reviewer 1 Report

Dear authors and editor,

Thank you for the opportunity to revise the manuscript entitled “Radiographic analysis of Grammont-style and lateralized reverse shoulder arthroplasty in gleno-humeral osteoarthritis”. The aim of this paper is interesting and will resume and explain radiographic parameters used in planning and follow up of reverse shoulder arthroplasty. It is well written and exposes probably more as a book chapter than a review of the literature, in a non-negative sense, the different steps to be followed before and after the surgery.

In my opinion the paper should be accepted as it is regardless of the special issue considered.

Author Response

Dear Reviewer,

Thank you for your thoughtful and positive feedback on our paper. We sincerely appreciate your kind words and the time you've dedicated to reviewing our work. We are pleased to hear that you found our research to be of high quality.

Your encouragement and support are invaluable to us.

Best regards,

Reviewer 2 Report

The article focuses on the analysis of X-ray images to compare Grammont-style and lateralized reverse shoulder arthroplasty (RSA) in patients with glenohumeral osteoarthritis. This paper is well-written and organized. However, it is recommended that the authors address the following changes before publication.

Reviewer Comments:

1.      Support the background section with scientific literature. From lines 34 to 43 authors have described several features of reverse prosthesis, and it is supported by an annual report. Please add a relevant scientific citation with each fact that has been described.

·      Similarly, in lines 54 – 55 please add a source of the fact.

“Although symptomatic GHOA is not as common as osteoarthritis of the hip and knee joints, it can be just as debilitating due to the functional importance of the upper limbs.”

·      And lines 66 – 67 “Disease progression is, however, typically affected by a combination of genetic, behavioural, and environmental factors.”

2.      Cite the source of Figures 2, 4, 5, 7, 8, 9, 10, and 11 in the caption, just like Figure 1.

3.      In Figure 7, remove the underlined text.

4.      In section 4, how are glenoid component height, tilt, scapular notching, radiolucent lines, and humeral component alignment crucial for monitoring the success and stability of the implant?

5.      In section 5, what radiographic parameters were used to measure lateralization, and please clarify the concept of "biologic lateralization"? Also, how does lateralization impact RSA stability?

6.      What biomechanical advantages does lateralization provide in RSA?

7.      Please check the manuscript for English grammatical structure, the following sentence needs to be rewritten.

“The biomechanical principles on which Grammont built its innovative reverse shoulder arthroplasty were four [1,44,45]:”

Add the conclusion section, including the limitation and the future scope.

Minor editing of English language required

Reviewer 3 Report

Review

 Title:” Radiographic analysis of Grammont-style and lateralized re- 2 verse shoulder arthroplasty in gleno-humeral osteoarthritis”

This study is a review article to compare Radiographic analysis of Grammont-style and lateralized reverse shoulder arthroplasty in gleno-humeral osteoarthritis.

It is well described article with proper method and conclusion. It is acceptable

Title

Good

Abstract

Good

Background

Good

Gleno-humeral deformity in primary osteoarthritis and cuff tear arthropathy

Good

Biomechanics of reverse shoulder arthroplasty

Good.

Radiographic evaluation of Grammont-style reverse arthroplasty

Good.

Radiographic features of lateralized reverse arthroplasty

Good.

References

Good.

Author Response

(The authors gave the same response as above.)
